# Time Difference of Arrival Passive Localization Sensor Selection Method Based on Tabu Search

**DOI:** 10.3390/s20226547

**Published:** 2020-11-16

**Authors:** Qian Li, Baixiao Chen, Minglei Yang

**Affiliations:** 1National Laboratory of Radar Signal Processing, Xidian University, Xi’an 710071, China; qli_10@stu.xidian.edu.cn (Q.L.); mlyang@xidian.edu.cn (M.Y.); 2Collaborative Innovation Center of Information Sensing and Understanding, Xidian University, Xi’an 710071, China

**Keywords:** passive localization, time difference of arrival (TDOA), sensor selection optimization, constrained total least-squares (CTLS), tabu search

## Abstract

This paper proposes a time difference of arrival (TDOA) passive positioning sensor selection method based on tabu search to balance the relationship between the positioning accuracy of the sensor network and system consumption. First, the passive time difference positioning model, taking into account the sensor position errors, is considered. Then, an approximate closed-form constrained total least-squares (CTLS) solution and a covariance matrix of the positioning error are provided. By introducing a Boolean selection vector, the sensor selection problem is transformed into an optimization problem that minimizes the trace of the positioning error covariance matrix. Thereafter, the tabu search method is employed to solve the transformed sensor selection problem. The simulation results show that the performance of the proposed sensor optimization method considerably approximates that of the exhaustive search method. Moreover, it can significantly reduce the running time and improve the timeliness of the algorithm.

## 1. Introduction

An essential feature in sonar, radar, environment monitoring rescue work, and next generation cellular phones is the capability of passively locating a source. The passive localization method does not emit a signal by itself, and only locates and tracks the source based on the signal radiated by the source, which has stronger concealment and survivability than the active positioning method. Several fundamental approaches for implementing a wireless location system, including those based on received signal strength (RSS) [1,2], angle of arrival (AOA) [3,4], time of arrival (TOA) [5,6], time difference of arrival (TDOA) [7,8,9,10], and frequency difference of arrival (FDOA), have been established. The common technique employed for stationary sources are the measurement of TDOAs using a number of spatially separated sensors.

In sensor networks, dense sensor deployment may introduce redundancy in the coverage area. Given a large sensor network, if all the sensors in the sensor network are activated, this is undoubtedly very inefficient. Since sensors far away from the sources do not contribute appreciably to the positioning performance, they consume energy and communication bandwidth of the sensor [11]. When the source moves, the combination of the most suitable positioning sensors before the source changed its location may no longer be optimal [12,13]. To attain the global optimization of positioning performance, it is necessary to exclude certain sensors from the original combination and introduce new sensors that are effective for positioning. Therefore, the optimal sensor network layout in the current situation must be selected according to the various positions of the source.

The problem of sensor selection is proposed in order to balance the positioning performance of the source and target and the loss of the system. Sensor selection is an integer programming problem with combinatorial complexity, making it problematic for large-scale sensor networks. To reduce this computational complexity, two concepts for sensor selection were introduced in recent years.

In the first concept, sparsity-promoting techniques have been proposed for sensor selection with the aim of minimizing the number of selected sensors and source state estimation error [14,15,16]. In the second concept, the number of sensors to be selected is predetermined, and the optimum subset of sensors is selected to maximize the estimated system performance. In the TDOA passive positioning scenario, the position of the source in two-dimensional space can be solved using four sensors. Accordingly, the influence of the second concept on the sensor selection scheme is mainly considered. Using the second concept, a convex relaxation procedure was developed in [17], and the sensor selection problem, formulated using linear measurement models, was solved via convex optimization. Considering the correlated measurement error [18], the constraints of the original problem were relaxed, the convex optimization was transformed into a standard semi-definite program (SDP), and the sensor selection problem was solved using a greedy algorithm. Based on the nonlinear estimation system, a Boolean selection vector was introduced [19], and an optimization problem was presented to minimize the number of selected sensors within the constraint of the preferred estimation accuracy. However, because the number of measurements is less than the number of sensors, the TDOA measurements differ from those of the universal nonlinear models. Similarly, in [20], the linearization of nonlinear function using the Taylor approximation technique was implemented, and a greedy algorithm based on two cost functions was proposed. However, the resulting performance was worse than that of the convex optimization method. In [21], a sensor selection method based on the TDOA positioning scheme using the weighted least squares algorithm for transforming the nonlinear equations into a pseudo-linear equation system was proposed. By introducing a Boolean selection vector, a non-convex sensor selection problem is formulated. When the non-convex constraint is relaxed, the non-convex problem can be effectively solved.

The main contributions of this paper are reflected in the two following points. First, when the number of available positioning sensors is given, a TDOA passive positioning sensor selection method based on the tabu algorithm is implemented, and the trace of the covariance matrix of the positioning error is used to evaluate the performance of the TDOA positioning system. In practical applications, the position of each sensor is expected to have a specific deviation. Accordingly, the approximate solution is obtained using the method of constrained total least squares (CTLS) considering the sensor position error. Then, based on this method, the positioning error covariance matrix is derived. Second, a Boolean selection vector is introduced, a sensor selection problem is formulated, the trace of the positioning error covariance matrix as the objective function is determined, and the tabu search algorithm is employed to solve the above problem. On the premise of ensuring the positioning accuracy, the calculation time of the algorithm is significantly reduced, and the timeliness of the algorithm is improved.

The remainder of this article is arranged as follows. In Section 2, the CTLS location method considering the sensor position error is introduced, and the derivation of the covariance matrix of the source location error is presented. In Section 3, a TDOA passive location sensor selection method based on the tabu search is proposed. To evaluate the performance of the proposed algorithm, the simulation results are discussed in Section 4. The conclusions are summarized in Section 5.

Notations: The superscript T denotes the transpose. ${\left(\cdot \right)}^{-1}$ and ${\left(\cdot \right)}^{+}$ represent the inverse and Moore-Penrose pseudoinverse, respectively. diag(a) denotes a diagonal matrix by taking the given vector a as the main diagonal. ${\left\|x\right\|}_{2}$ denotes the $l_{2}$-norm of a vector x. $I_{N}$ is the identity matrix with size N. $0_{N}$ is the zero matrix with size N. tr(⋅) is the matrix trace operator. E(⋅) denotes the expectation operation. ⌊m⌋ represents the largest positive integer not exceeding m. The symbol ⊗ is used to denote the Kronecker product.

## 2. TDOA-Based Location Problem Model

### 2.1. TDOA-Based Localization Problem

For convenience, a two-dimensional (2-D) scenario is considered where a network of M passive sensors collaborates to localize a stationary point source with an unknown position $v_{0}={\left[x_{t},y_{t}\right]}^{T}$. The coordinates of the passive sensors are $s_{i}={\left[x_{i},y_{i}\right]}^{T},i=1,2,\ldots ,M$. The first sensor is assumed as the reference sensor and the distance from the source to the sensor i is $r_{i}$. Through the measurement of the TDOA, the basic measurement equation is derived as follows
(1)$$r_{i1}=r_{i}-r_{1}=\sqrt{{\left(x_{t}-x_{i}\right)}^{2}+{\left(y_{t}-y_{i}\right)}^{2}}-\sqrt{{\left(x_{t}-x_{1}\right)}^{2}+{\left(y_{t}-y_{1}\right)}^{2}}=c\cdot \Delta t_{i1},\;i=2,3,\ldots ,M.$$
where $r_{i1}$ indicates the difference between the distance from the source to the *i*-th sensor and the distance from the source to the first sensor. $\Delta t_{i1}$ is the true value of the TDOA measurement between the first sensor and sensor i. c is the speed of light. Squaring both sides of (1) yields the following
(2)$$(x_{i}-x_{1})(x_{t}-x_{1})+(y_{i}-y_{1})(y_{t}-y_{1})+r_{i1}\cdot r_{1}=\frac{1}{2}({\left(x_{i}-x_{1}\right)}^{2}+{\left(y_{i}-y_{1}\right)}^{2}-r_{i1}{}^{2}).$$

Formulating (2) in matrix form yields
(3)Aθ=b,
where
$$A=\left[\begin{array}{ccc} x_{2}-x_{1} & y_{2}-y_{1} & r_{21}\\ x_{3}-x_{1} & y_{3}-y_{1} & r_{31}\\ \ldots & \ldots & \ldots \\ x_{M}-x_{1} & y_{M}-y_{1} & r_{M1} \end{array}\right],\;\theta =\left[\begin{array}{c} x_{t}-x_{1}\\ y_{t}-y_{1}\\ r_{1} \end{array}\right],\;b=\frac{1}{2}\left[\begin{array}{c} {\left(x_{2}-x_{1}\right)}^{2}+{\left(y_{2}-y_{1}\right)}^{2}-r_{21}^{2}\\ {\left(x_{3}-x_{1}\right)}^{2}+{\left(y_{3}-y_{1}\right)}^{2}-r_{31}^{2}\\ \ldots \\ {\left(x_{M}-x_{1}\right)}^{2}+{\left(y_{M}-y_{1}\right)}^{2}-r_{M1}^{2} \end{array}\right]$$

According to the principle of least squares, the solution of the source position can be obtained as
(4)$$\theta_{LS}=\underset{\hat{\theta }}{\arg\;\;\min}{\left(A\hat{\theta }-b\right)}^{T}(A\hat{\theta }-b)={\left(A^{T}A\right)}^{-1}A^{T}b.$$

### 2.2. Constrained Total Least Squares Algorithm under Error Conditions

It is worth noting that, in practice, both **A** and **b** in (3) have errors, which do not conform to the assumption of the least squares method. The authors in [22] used the CTLS algorithm to solve the problem to reduce the impact of errors. Below we derive the CTLS solution of (3) when considering TDOA measurement error and sensor position error.

In (3), the matrix A and vector b usually contain error components. Thus, Equation (3) can be rewritten as
(5)$$(A_{0}+\Delta A)\theta =(b_{0}+\Delta b),$$
where $A_{0}$, $b_{0}$ represent true values, and ΔA, Δb represent the error components.

Assuming that each sensor is independent and identically distributed, the position error in the *i*-th sensor $n_{x_{i}}$, $n_{y_{i}}$ obey the Gaussian distribution whose mean and the standard deviation are set as 0 and $\sigma_{s}$, respectively. The TDOA measurement error $n_{r_{i1}}$ obeys the Gaussian distribution whose mean and standard deviation are set as 0, $\sigma_{t}$, respectively. Then
(6)$${\hat{x}}_{i}=x_{i}+n_{x_{i}},{\hat{y}}_{i}=y_{i}+n_{y_{i}},i=2,3,\ldots ,M.$$
where ${\hat{x}}_{i}$, ${\hat{y}}_{i}$ represents the coordinate value with error, and $x_{i}$, $y_{i}$ represents the true coordinate value without any error.

It is assumed that, in $A=\left[\begin{array}{cc} A_{2} & A_{3} \end{array}\right]$, $A_{2}$ consists of the first two columns of A, and $A_{3}$ consists of the last column of A [22]. The following is obtained
(7)$$\Delta A=\left[\begin{array}{cc} \Delta A_{2} & \Delta A_{3} \end{array}\right],$$
where
$$\Delta A_{2}=\left[\begin{array}{cc} \Delta (x_{2}-x_{1}) & \Delta (y_{2}-y_{1})\\ \Delta (x_{3}-x_{1}) & \Delta (y_{3}-y_{1})\\ \ldots & \ldots \\ \Delta (x_{M}-x_{1}) & \Delta (y_{M}-y_{1}) \end{array}\right],\;\Delta A_{3}={\left[\Delta r_{21},\Delta r_{31},\ldots ,\Delta r_{M1}\right]}^{T}.$$

The elements of (*i* − 1)-th (i=2,3,…,M) row in matrix $\Delta A_{2}$ can be represented according to the following
(8)$$\Delta ({\hat{x}}_{i}-x_{1})=({\hat{x}}_{i}-x_{1})-(x_{i}-x_{1})=x_{i}+n_{x_{i}}-x_{i}=n_{x_{i}},$$
(9)$$\Delta ({\hat{y}}_{i}-y_{1})=({\hat{y}}_{i}-y_{1})-(y_{i}-y_{1})=y_{i}+n_{y_{i}}-y_{i}=n_{y_{i}}.$$

The elements in $\Delta A_{3}$ can be expressed as
(10)$$\Delta r_{i1}=({\hat{r}}_{i1}-r_{i1})=n_{r_{i1}},$$

Similarly, we can obtain
(11)$$\Delta b={\left[\Delta b_{2},\Delta b_{3},\ldots ,\Delta b_{M}\right]}^{T}$$

The elements in Δb can be expressed as follows
(12)$$\begin{array}{l} \Delta b_{i}={\hat{b}}_{i}-b_{i}=\frac{1}{2}\left[{\left(x_{i}+n_{x_{i}}-x_{1}\right)}^{2}+{\left(y_{i}+n_{y_{i}}-y_{1}\right)}^{2}-{\left(r_{i1}+n_{r_{i1}}\right)}^{2}\right]-\frac{1}{2}\left[{\left(x_{i}-x_{1}\right)}^{2}+{\left(y_{i}-y_{1}\right)}^{2}-r_{i1}{}^{2}\right]\\ \;\;\;\;\;\;\;=(x_{i}-x_{1})n_{x_{i}}+\frac{1}{2}n_{x_{i}}^{2}+(y_{i}-y_{1})n_{y_{i}}+\frac{1}{2}n_{y_{i}}^{2}-r_{i1}n_{r_{i1}}-\frac{1}{2}n_{r_{i1}}^{2}. \end{array}$$

Ignoring the influence of the quadratic term of (12) yields
(13)$$\Delta b_{i}\;=(x_{i}-x_{1})n_{x_{i}}+(y_{i}-y_{1})n_{y_{i}}-r_{i1}n_{r_{i1}}.$$

The error vector is denoted as
(14)$$\eta ={\left[\begin{array}{ccc} n_{x} & n_{y} & n_{t} \end{array}\right]}^{T},$$
where
$$n_{x}=[\begin{array}{cccc} n_{x_{2}} & n_{x_{3}} & \ldots & n_{x_{M}} \end{array}],\;n_{y}=[\begin{array}{cccc} n_{y_{2}} & n_{y_{3}} & \ldots & n_{y_{M}} \end{array}],\;n_{t}=[\begin{array}{cccc} n_{r_{21}} & n_{r_{31}} & \ldots & n_{r_{M1}} \end{array}].$$

Substituting (11) into (8) yields
(15)$$\{\begin{array}{c} \Delta A=[\begin{array}{ccc} G_{1}\eta & G_{2}\eta & G_{3}\eta \end{array}]\\ \Delta b=G_{4}\eta \end{array},$$
where
$$G_{1}=\left[\begin{array}{ccc} I_{M-1} & 0_{M-1} & 0_{M-1} \end{array}\right],\;G_{2}=\left[\begin{array}{ccc} 0_{M-1} & I_{M-1} & 0_{M-1} \end{array}\right],\;G_{3}=\left[\begin{array}{ccc} 0_{M-1} & 0_{M-1} & I_{M-1} \end{array}\right],\;\begin{array}{l} G_{4}=[\mathrm{diag}(x_{2}-x_{1},x_{3}-x_{1},\ldots ,x_{M}-x_{1}),\\ \;\;\;\;\;\;\;\;\;\;\;\mathrm{diag}(y_{2}-y_{1},y_{3}-y_{1},\ldots ,y_{M}-y_{1}),\\ \;\;\;\;\;\;\;\;\;\;\;\mathrm{diag}(-r_{21},-r_{31},\ldots ,-r_{M1})]. \end{array}$$

The position error of the observation station and time difference measurement error are independent of each other, the autocorrelation matrix of the error vector is $R_{\eta }$. The covariance matrix of each element in η is different, hence, η can be whitened, the white error vector that obeys the Gaussian distribution after η whitening is e. To obtain $R_{\eta }=PP^{T}$, the Cholesky decomposition is implemented. Then, the process of error vector whitening is $e=P^{-1}\eta$. The relationship between the error vector η and the normal distribution white error vector e is η=Pe. Substituting this into (15), yields
(16)$$\{\begin{array}{c} \Delta A=[\begin{array}{ccc} G_{1}Pe & G_{2}Pe & G_{3}Pe \end{array}]\\ \Delta b=G_{4}Pe \end{array}$$

From (3) and (16), we obtain the following
(17)$$\begin{array}{ll} A\theta -b & =(A_{0}+\Delta A)\theta -(b_{0}+\Delta b)\\ & =\Delta A\theta -\Delta b\\ & =(x_{t}-x_{1})G_{1}Pe+(y_{t}-y_{1})G_{2}Pe+r_{1}G_{3}Pe-G_{4}Pe\\ & =GPe, \end{array}$$
where $G=(x_{t}-x_{1})G_{1}+(y_{t}-y_{1})G_{2}+r_{1}G_{3}-G_{4}$. The error vector e is given by
(18)$$e={\left(GP\right)}^{+}(A\theta -b),$$
where ${\left(GP\right)}^{+}={\left(GP\right)}^{T}{\left(GR_{\eta }G^{T}\right)}^{-1}$. Next, the $l_{2}$-norm of error vector e is minimized to obtain the objective function
(19)$$\{\begin{array}{c} \underset{\theta ,e}{\min}{\left\|e\right\|}_{2}^{2}\\ s.t.\;\;\;\;A\theta -b=GPe \end{array}.$$

According to (18) and (19), the objective function of the CTLS solution can be obtained as
(20)$$F(\theta )={\left({\left(GP\right)}^{+}(A\theta -b)\right)}^{T}({\left(GP\right)}^{+}(A\theta -b))={\left(A\theta -b\right)}^{T}{\left(GR_{\eta }G^{T}\right)}^{-1}(A\theta -b).$$

The objective function is a real nonlinear function with respect to θ, hence, it cannot be solved directly. Consequently, all relevant literature on the CTLS method reports the use of the Newton algorithm as the iterative solution. Moreover, because the iterative method necessitates the calculation of the Hess matrix and gradient vector of the objective function, multiple matrix inversions and multiplication operations are necessary, resulting in high computational complexity and longer solution time. The approximate closed-form solution without iteration is given below.

A necessary condition [23] to minimize F(θ) is ∂F(θ)/∂θ=0. Thus, the unconstrained objective function derivative of (20) is
(21)$$\frac{\partial F(\theta )}{\partial \theta }=2A^{T}{\left(GR_{\eta }G^{T}\right)}^{-1}(A\theta -b)+{\left(A\theta -b\right)}^{T}\frac{\partial {\left(GR_{\eta }G^{T}\right)}^{-1}}{\partial \theta }(A\theta -b)=0.$$

The second term can be written as
(22)$${\left(A\theta -b\right)}^{T}\frac{\partial {\left(GR_{\eta }G^{T}\right)}^{-1}}{\partial \theta }(A\theta -b)=e^{T}P^{T}G^{T}\frac{\partial {\left(GR_{\eta }G^{T}\right)}^{-1}}{\partial \theta }GPe.$$

Equation (22) is the second-order term of e containing Gaussian error. Ignoring the second-order term, Equation (21) can be approximately derived as
(23)$$A^{T}{\left(GR_{\eta }G^{T}\right)}^{-1}(A\theta -b)=0.$$

By solving the foregoing equation, the approximate closed-form solution of CTLS is
(24)$$\theta_{CTLS}\approx {\left[A^{T}{\left(GR_{\eta }G^{T}\right)}^{-1}A\right]}^{-1}A^{T}{\left(GR_{\eta }G^{T}\right)}^{-1}b.$$

### 2.3. Location Error Covariance Matrix

The intermediate variable $r_{1}$ is introduced in this section it can be expressed as
(25)$$r_{1}=\sqrt{{\left(x_{t}-x_{1}\right)}^{2}+{\left(y_{t}-y_{1}\right)}^{2}}={\left(v^{T}v\right)}^{\frac{1}{2}},$$
where $v={\left[x_{t}-x_{1},y_{t}-y_{1}\right]}^{T}$. Substituting (25) into (20) yields
(26)$$F(v)={\left(A_{2}v+A_{3}{\left(v^{T}v\right)}^{1/2}-b\right)}^{T}{\left(GR_{\eta }G^{T}\right)}^{-1}\cdot (A_{2}v+A_{3}{\left(v^{T}v\right)}^{1/2}-b).$$

For the convenience of the following description, it may be assumed
(27)$$\Omega =A_{2}v+A_{3}{\left(v^{T}v\right)}^{1/2}-b,$$
(28)$$\Theta =A_{2}^{T}+{\left(v^{T}v\right)}^{-1/2}vA_{3}^{T}.$$

Similar to the process used in (21), substituting (27) and (28) into (26) yields the following
(29)$$\frac{\partial F(v)}{\partial v}=2\Theta {\left(GR_{\eta }G^{T}\right)}^{-1}\Omega +\Omega^{T}\frac{\partial {\left(GR_{\eta }G^{T}\right)}^{-1}}{\partial v}\Omega ,$$

The second term in (29) contains the quadratic term of error. Ignoring the quadratic term can be obtained
(30)$$\frac{\partial F(v)}{\partial v}=2\Theta \mu .$$
where $\mu ={\left(GR_{\eta }G^{T}\right)}^{-1}\Omega$. Let the value of the above Equation (30) be 0, consequently,
(31)$$2\Theta \mu =2\Theta \cdot {\left(GR_{\eta }G^{T}\right)}^{-1}\Omega =0.$$

When an error exists between the TDOA and the sensor position, the perturbation method is employed to calculate the covariance matrix of the positioning error.
(32)$$\begin{array}{l} \Omega =(A_{20}+\Delta A_{2})(v+\Delta v)+(A_{30}+\Delta A_{3}){\left({\left(v+\Delta v\right)}^{T}(v+\Delta v)\right)}^{1/2}-(b_{0}+\Delta b)\\ \;\;\;=A_{20}\Delta v+\Delta A_{2}(v+\Delta v)+A_{30}\frac{v^{T}\Delta v}{{\left(v^{T}v\right)}^{1/2}}+(A_{30}+\Delta A_{3})\Delta v^{T}\Delta v+\Delta A_{3}{\left(v^{T}v\right)}^{1/2}+\Delta A_{3}\frac{v^{T}\Delta v}{{\left(v^{T}v\right)}^{1/2}}-\Delta b\\ \;\;\;\;\approx A_{20}\Delta v+A_{30}\frac{v^{T}\Delta v}{{\left(v^{T}v\right)}^{1/2}}+\Delta A_{2}v+\Delta A_{3}{\left(v^{T}v\right)}^{1/2}-\Delta b\\ \;\;\;\;=A_{20}\Delta v+A_{30}\frac{v^{T}\Delta v}{{\left(v^{T}v\right)}^{1/2}}+GPe, \end{array}$$
where the second equal sign in (32) is because
(33)$$\begin{array}{ll} A_{0}\theta -b_{0} & =\left[\begin{array}{cc} A_{20} & A_{30} \end{array}\right]\cdot \left[\begin{array}{c} v\\ {\left(v^{T}v\right)}^{1/2} \end{array}\right]-b_{0}\\ & \;\;\;=A_{20}v+A_{30}{\left(v^{T}v\right)}^{1/2}-b_{0}\\ & \;\;\;=0 \end{array}$$

Ignoring the perturbation terms higher than order 1 yields the following from (28)
(34)$$\begin{array}{l} \Theta ={\left(A_{20}+\Delta A_{2}\right)}^{T}+{\left({\left(v+\Delta v\right)}^{T}(v+\Delta v)\right)}^{-\frac{1}{2}}(v+\Delta v){\left(A_{30}+\Delta A_{3}\right)}^{T}\\ \;\;\;\;\approx A_{20}{}^{T}+{\left(v^{T}v\right)}^{-\frac{1}{2}}vA_{30}{}^{T}. \end{array}$$

Substituting (32) and (34) into (31) yields the following
(35)$$D^{T}{\left(GR_{\eta }G^{T}\right)}^{-1}(D\Delta v+GPe)=0,$$
where
$$D=A_{20}+A_{30}\frac{v^{T}}{{\left(v^{T}v\right)}^{1/2}}.$$

It follows from (35) that
(36)$$\Delta v=-{\left(D^{T}{\left(GR_{\eta }G^{T}\right)}^{-1}D\right)}^{-1}D^{T}{\left(GR_{\eta }G^{T}\right)}^{-1}GPe.$$

The covariance matrix of the location error is given by
(37)$$\begin{array}{l} E(\Delta v\Delta v^{T})={\left(D^{T}{\left(GR_{\eta }G^{T}\right)}^{-1}D\right)}^{-1}D^{T}{\left(GR_{\eta }G^{T}\right)}^{-1}\;\cdot GPE(ee^{T})P^{T}G^{T}{\left(GR_{\eta }G^{T}\right)}^{-1}\\ \;\;\;\;\;\;\;\;\;\;\;\;\;\;\;\;\;\;\;\;\;\;\;\;\;\;\;\cdot D{\left(D^{T}{\left(GR_{\eta }G^{T}\right)}^{-1}D\right)}^{-1}\\ \;\;\;\;\;\;\;\;\;\;\;\;\;\;\;\;\;\;\;={\left(D^{T}{\left(GR_{\eta }G^{T}\right)}^{-1}D\right)}^{-1}. \end{array}$$

## 3. Proposed Sensor Selection Optimization Method

The previous section discussed the CTLS method of the TDOA passive positioning with sensor error. In response to the actual requirements for optimal sensor selection, this paper proposes a sensor selection method based on tabu search.

Depending on the actual requirement in practice, the sensor selection objective equations for different optimization goals differ. Consider a network in which M available sensors and used K (K < M) positioning sensors are known. A technique for selecting K to minimize positioning errors is presented in this paper. The TDOA measurement error and the sensor position error obey the Gaussian distribution with zero means and variances $\sigma_{t}^{2}$, $\sigma_{s}^{2}$, respectively. When they are in a small range, the performance of the CTLS method can reach Cramér–Rao lower bound (CRLB). Accordingly, this paper uses the covariance matrix of the positioning error of the positioning source (37) as the objective function to identify the optimal sensor.

A Boolean vector, $w={\left[w_{1},w_{2},\ldots ,w_{M}\right]}^{T}$, where $w_{i}\in \left\{0,1\right\}$, is introduced for determining whether a particular sensor is selected. If the *i*-th element in w is 1, then, the *i*-th sensor is selected to be a part of the optimal subset [19]. In the TDOA positioning scenario, the sensor receiving the highest signal strength is regarded as the reference sensor. After all the sensors have received the signal, since the reference sensor has been given, the Boolean vector is transformed as
(38)$$w={\left[w_{2},w_{3},\ldots ,w_{M}\right]}^{T},w_{i}\in \left\{0,1\right\}$$

This means that K-1 sensors are selected from the remaining M-1 sensors. The definition matrix $\Phi_{w}$, which is the sub-matrix of diag(w) after removing the rows of unselected sensors, and the relationship between $\Phi_{w}$ and w is given by
(39)$$\Phi_{w}\Phi_{w}{}^{T}=I_{{\left\|w\right\|}_{1}},\Phi_{w}{}^{T}\Phi_{w}=diag(w)$$

After sensor selection, the inverse matrix of the covariance matrix of the positioning source position error is expressed as
(40)$$\begin{array}{l} J_{w}=E{\left(\Delta v_{w}\Delta v_{w}^{T}\right)}^{-1}=D_{w}^{T}{\left(G_{w}R_{\eta_{w}}G_{w}^{T}\right)}^{-1}D_{w}\\ \;\;\;\;\;\;\;\;=D^{T}\Phi_{w}^{T}{\left(\Phi_{w}G\Psi \Psi^{T}R_{\eta }\Psi \Psi^{T}G^{T}\Phi_{w}^{T}\right)}^{-1}\Phi_{w}D \end{array}$$
where $\Psi =I_{3}⊗\Phi_{w}^{T}$.

From this, the sub-matrix of the row where the unselected sensor is removed from the $D_{w}$ expression matrix D. $G_{w}$ represents the matrix filtered from the system matrix G using the Boolean vector w.

In this paper, the A-optimality model in [19] is selected as the constraint problem. The trace of the covariance matrix of the positioning error is used as the objective function. Thus, the sensor selection problem can be expressed as
(41)$$\left\{ \begin{array}{l} \min\;\;{\mathrm{tr}(J}_{w}^{-1})\\ s.t.\;\;1^{T}w=K-1\\ \;\;\;\;\;\;\;\;\;\;\;w_{i}\in \left\{0,1\right\}\;\;\;\;\;i=2,3,\ldots ,M \end{array}\right.$$

To solve the foregoing, this paper employs the tabu search method to obtain the optimal subset. The tabu search algorithm [24] is an iterative search algorithm. It was first proposed by Professor Fred Glover, a member of the American Academy of Engineering, and further defined and developed [25,26]. Its core idea is to introduce a tabu table to record the local optimum points that have been searched. In the succeeding search, the information in the tabu table is either selectively searched or not searched to jump out of the local optimum and achieve global optimization. Thus far, tabu search algorithms have achieved remarkable success in combinatorial optimization, production scheduling, machine learning, and neural networks [27,28].

The optimization process of modern heuristic algorithms is a “neighborhood search” structure. The algorithm starts from an initial solution and generates several neighborhood solutions through neighborhood functions under the control of the key parameters of the algorithm, and updates the current state according to the acceptance criteria. Repeat the above search steps until the convergence criterion of the algorithm is satisfied, and finally get the optimization result of the problem. As a heuristic algorithm, the tabu search method is distinguished from other modern heuristic algorithms in that it uses memory to guide the search process of the algorithm. It is a simulation of human intelligence process and a manifestation of artificial intelligence. It can be seen that, when using a heuristic algorithm to solve a problem, it is generally necessary to determine the representation method of the solution, its evaluation criteria, neighborhood operation rules, and termination rules. These factors directly affect the speed of the algorithm and the quality of the solution. When using the tabu search algorithm, it is also necessary to design the tabu object, length, and candidate set:(a)Representation method of solution: use vector w to indicate whether each sensor is selected.(b)Solution evaluation criterion: use the trace of the covariance matrix of the source positioning error.(c)Operation rule of neighborhood: Interchange the state of any selected sensor and any unselected sensor in the current solution to form a set of neighborhoods. Thus, the number of neighborhood solutions is
(42)$$C_{a}=C_{K-1}^{1}\cdot C_{M-K}^{1}=(K-1)\cdot (M-K)$$(d)Termination criterion: when the number of iterations reaches the maximum $Iter_{\max}=M$ (where M is the size of the sensor network), the algorithm terminates.(e)Selection of tabu objects: Select tabu objects as simple solution changes. A simple solution change can reduce the forbidden range and expand the search range.(f)Tabu length: the length of the tabu table is selected as $TabuL=\left\lfloor sqrt(C_{a})\right\rfloor$.(g)Determination of candidate set: randomly select several neighborhood solutions as candidate sets in the neighborhood of the current solution.

After determining the foregoing parameters, the steps of the TDOA passive positioning sensor selection algorithm based on tabu search are as follows:Step 1. Initialize the parameters and initial sensor selection solution and set the tabu table as an empty set.Step 2. Assess whether the algorithm termination condition (d) is satisfied; if so, then, end the algorithm and output the result; otherwise, proceed to the following steps.Step 3. Use the current solution to generate $C_{a}$ neighborhood solutions and candidate solutions.Step 4. Evaluate the tabu attribute of each object corresponding to the candidate solution. Select the minimum value of the objective function corresponding to the non-tabu object in the candidate solution set as the new current solution. Replace the earliest tabu object element that has been listed in the tabu table with the corresponding tabu object concurrently (update the tabu table).Step 5. Return to Step 2.

## 4. Simulation Results

In this section, the simulation results of the sensor selection method based on the tabu search for the TDOA passive positioning are presented. Moreover, the result of the proposed algorithm is compared with those of the algorithm used in [18], the algorithm used in [21], the nearest distance method, and the exhaustive search method as the reference. The exhaustive search method requires an exhaustive combination of sub-sensors, the technique has a high computational complexity and poor timeliness. However, an optimal combination of positioning sensors can be determined as the optimal standard for comparing various sensor optimization methods. The number of Monte Carlo runs L=500. The root mean square error and the average calculation time of 100 Monte Carlo experiments as indicators to measure the positioning performance are used
(43)$$\mathrm{RMSE}=\sqrt{\sum_{i=1}^{L}{\left\|v^{\left(i\right)}-v_{0}\right\|}^{2}/L}$$
(44)$$t=\frac{100}{L}\cdot \sum_{i=1}^{L}t^{\left(i\right)}$$

In the simulation, the average computing time was calculated using MATLAB2017a software on a Win7 64-bit desktop computer with i7-4790 clocked at 3.60 GHz and a 16 GB memory capacity.

### 4.1. Influence of TDOA Measurement Errors

Let us assume that the source position is (1000, 1200) m. The positioning network contains M = 10 sensors, which are randomly distributed in an area with a radius of 3000 m. The coordinate of the origin is (0, 0) m, and a combination of K = 4 positioning sensors is selected. The positions of the 10 sensors are (655, 1020), (1050, 2791), (−1550, −1281), (−657, 2636), (−478, 1666), (1806, 1302), (1461, 580), (−172, −2732), (986, −1075), and (−1024, −291) m. The measurement error of the site obeys the Gaussian distribution with a mean value of 0 and a standard deviation of 5 m. Figure 1 shows the schematic of the positioning network distribution.

The algorithm in [18], the algorithm in [21], the exhaustive search method, and proposed method are employed to solve the sensor selection problem (38). A schematic of the sensor selection results of each method when the standard deviations of the TDOA measurement error and sensor position error are 15 m and 5 m, respectively, is shown in Figure 2. It is worth noting that the algorithm in [18] and proposed algorithm give the same sensor selection results.

It can be observed in Figure 3 that the positioning accuracy of several different sensor selection methods can be compared as the TDOA measurement error increases. The algorithm proposed in this paper has the same performance as the method presented in [18]. This is because the sensor selection vector w given by the two methods is the same when solving the optimization problem of (41); consequently, their RMSE are also the same. The performance of the proposed algorithm better approximates that of the exhaustive search method, followed by that of the nearest distance method, and then that of the algorithm in presented in [21].

Figure 4 shows the variation curves of the RMSE of several algorithms with TDOA measurement error, and the error bars are shown in the figure. As can be seen from Figure 4, as the TDOA measurement error increases, the variation range of the error bar also increases. Since a positioning method is used, the difference between the error bars between the methods is not very significant.

Table 1 shows that the number of times of performing 100 Monte Carlo experiments, the average running time required for each method and the root mean square (RMS) deviation of running time when the standard deviation of TDOA measurement error is 15 m and the standard deviation of sensor position error is 5 m.

The CVX convex optimization toolbox is employed to solve the optimizing sensor problem in [18,21]; consequently, its running time is significantly increased. The calculation time of the method proposed in this paper is considerably less than those of the other methods. Compared with the algorithm in [18], the average running time of the algorithm is greatly reduced under the same positioning accuracy. According to the foregoing simulations, the sensor selection method for the TDOA passive positioning proposed in this paper significantly reduces the calculation time, improves the real-time calculation performance, and maintains a high positioning accuracy.

### 4.2. Influence of the Positions of the Source

In this section, in order to consider the impact on the performance of each method when the source is at different locations, we assumed that the remaining conditions are the same as those in Section 4.1, the locations of the sources are randomly distributed in a circle with a center of (0, 0) m and a radius of 2000 m. The number of Monte Carlo runs L=…. The performance analysis of each method when the source is located at different locations is given below. Then two special cases are given; namely the Geometric Dilution of Precision (GDOP) distribution when the source are located at (1000, 1200) m, (0, −1200) m, respectively. The calculation formula of GDOP is as follows
(45)$$\mathrm{GDOP}=\sqrt{tr(J_{w}^{-1})}$$

When the sensors are evenly distributed in a circular area, the variation curve of RMSE and TDOA measurement error of positioning error is shown in Figure 5. As the measurement error of TDOA increases, the RMSE of each method also increases. Among them, the algorithm proposed in this paper and the algorithm in [18] have better positioning performance. It is worth noting that this is because the location of the source in Section 4.1 is a special case of the situation in Section 4.2. Therefore, the conclusions reached by the two are consistent.

It can be observed from Figure 6 that, when the source is located at different positions, the selection results of the sensors are also different. It is worth noting that, when the sensor adopts the distribution scheme in Figure 6a, the positioning effect at point (0, −1200) m is not very good; that is, when the source is located at (0, −1200) m, the sensor that adopts the distribution method is not reasonable, and the sensor distribution scheme in Figure 6b should be adopted. Therefore, for sources at different locations, different sensor distribution methods should be adopted to achieve the best positioning performance.

### 4.3. Influence of the Sensor Position Errors

The distribution of the sensors is the same as Section 4.1, the standard deviation of TDOA measurement error is 5 m. The simulation is performed with the sensor position errors ranging from 1 m to 10 m. The change curve of RMSE and the sensor position errors of each algorithm are given in Figure 7.

It can be seen from Figure 7 that the algorithm proposed in this paper still demonstrates a better performance compared with the other methods. As the sensor position errors increase, the RMSE of several algorithms also increases; the algorithm proposed and the algorithm in [18] produce the minimum RMSE, followed by the nearest distance method using the algorithm in [21]. Since the same sensor selection vector w is solved, the RMSE of the proposed algorithm and algorithm in [18] are the same.

### 4.4. Influence of the Number of Sensors

The source position is (1000, 1200) m. The positioning network contains M = 20 sensors, which are randomly distributed in an area with a radius of 3000 m. The coordinate of the origin is (0, 0) m, and a combination of K = 4 positioning sensors is selected. The positions of the 20 sensors are (290, −756), (443, 2342), (−534, −1438), (−285, 1358), (−333, −2711), (−2592, 1507), (1321, −59), (−31, −2959), (−2717, −619), (−893, 1594), (−2536, 860), (2010, 1752), (−2976, −64), (580, 6), (2446, −1418), (−366, 141), (922, −1950], (−528, −1017), (2465, 670), (−1039, 2382) m. The standard deviation of the sensor position error is 5 m.

When the number of the sensor is M = 20, the variation curve of RMSE with TDOA measurement errors is shown in Figure 8. The method proposed in this paper can still have better positioning performance. Figure 9 shows the variation curves of the RMSE of several algorithms with TDOA measurement error, and the error bars are shown in the figure. As can be seen from Figure 9, we get the same conclusion as in Section 4.1.

Table 2 shows that the number of times of performing 100 Monte Carlo experiments, the average running time required for each method and the RMS deviation of running time when the standard deviation of TDOA measurement error is 10 m. After comparing Table 1 and Table 2, it can be seen that as the number of sensors to be selected increases, and the average running time of the exhaustive search method greatly increases. The calculation efficiency of the method proposed in this paper is much better than the other methods, which is consistent with the previous conclusions.

## 5. Conclusions

The sensor selection in passive location is a process full of uncertainty, and the result of its selection depends on the location algorithm, the measure data, the objective function and other factors, which brings difficulties to sensor selection and layout. Based on the introduction of the CTLS algorithm for TDOA passive positioning under the condition of site error, this paper gives the positioning error covariance matrix under this algorithm, and transforms the sensor selection problem that minimizes the objective function under the condition of a fixed number of sensors. To solve the problem, a TDOA passive location sensor selection method based on tabu search algorithm is proposed. We have calculated the optimal solution to the above problem by establishing a tabu table and operating rules in the neighborhood. In the simulation experiment, the positioning performance and the average running time of the algorithm under different parameter conditions are compared, including the influence of TDOA measurement error, sensor position error, and the size of the sensor network. The GDOP distribution map when the source is located in different positions is given. Then, we compare the positioning performance in different scenarios and the average running time of the algorithm. The simulation results show that the method proposed in this paper can be close to the exhaustive search method in positioning accuracy, greatly reducing the running time of the algorithm, and overcoming the shortcomings of traditional algorithms of high computational complexity and poor timeliness. Meanwhile, the proposed method is also helpful in improving the selection result by adjusting sensors, which is more reasonable and convincing. In summary, the algorithm proposed in this article has the following advantages: (1) This algorithm takes into account the influence of the position error of the sensor on the final result. (2) This algorithm can greatly shorten the calculation time while ensuring high positioning accuracy. (3) This algorithm is simple and easily operated. Therefore, the method can be applied to the selection and evaluation of sensors to perform TDOA-based source positioning, improve the overall performance of the positioning method, and facilitate the layout of sensors.

## Figures and Tables

**Figure 1 sensors-20-06547-f001:**
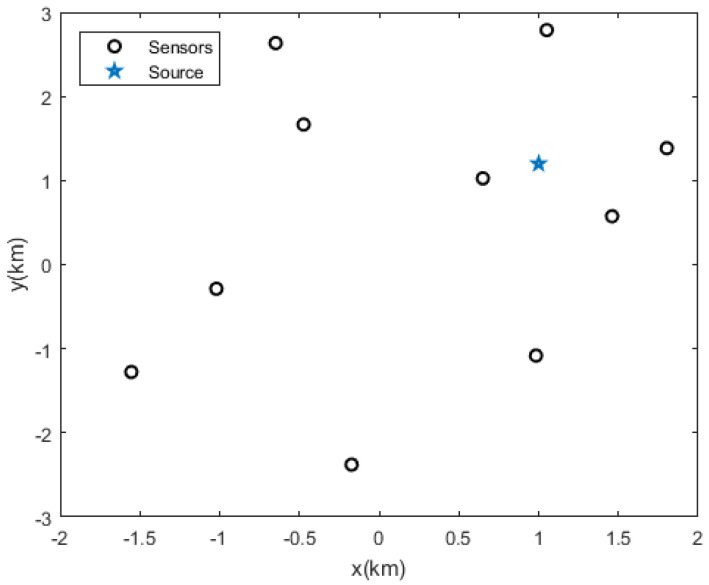
Distribution of sensor network (taking the source position in (1000, 1200) m as an example).

**Figure 2 sensors-20-06547-f002:**
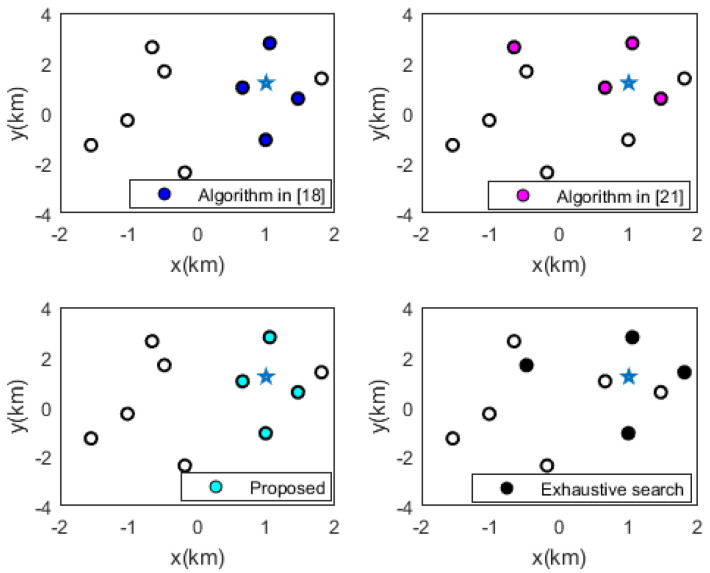
Optimal results of sensor selection using different methods (taking the source position in (1000, 1200) m as an example).

**Figure 3 sensors-20-06547-f003:**
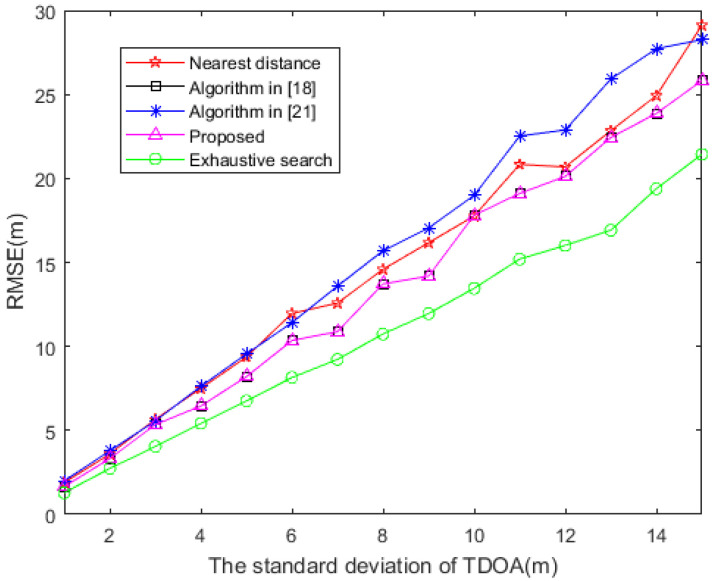
RMSE based on distinct TDOA measurement error (M = 10).

**Figure 4 sensors-20-06547-f004:**
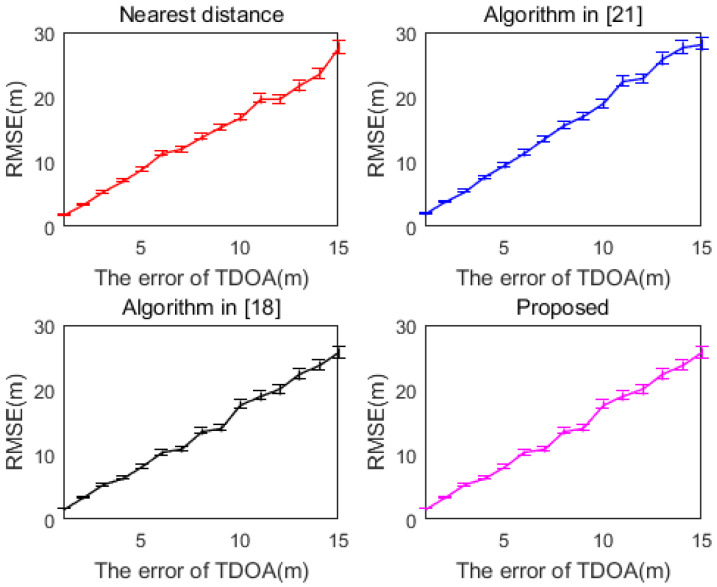
RMSE curve of each method with error bars (M = 10).

**Figure 5 sensors-20-06547-f005:**
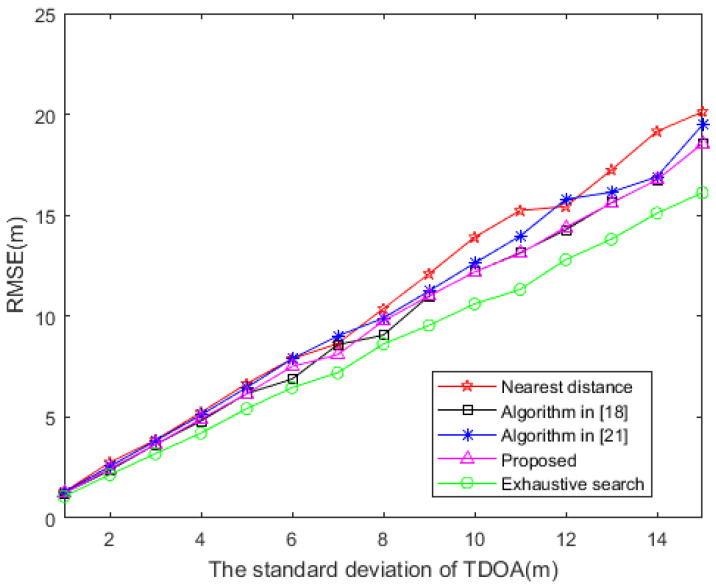
RMSE based on distinct TDOA measurement error. (The source is evenly distributed in a circular area).

**Figure 6 sensors-20-06547-f006:**
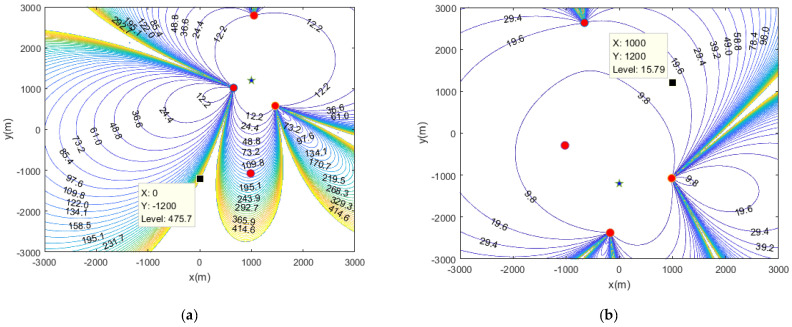
GDOP distribution when the location of the source is (1000, 1200) m (**a**), the location is (0, −1200) m (**b**).

**Figure 7 sensors-20-06547-f007:**
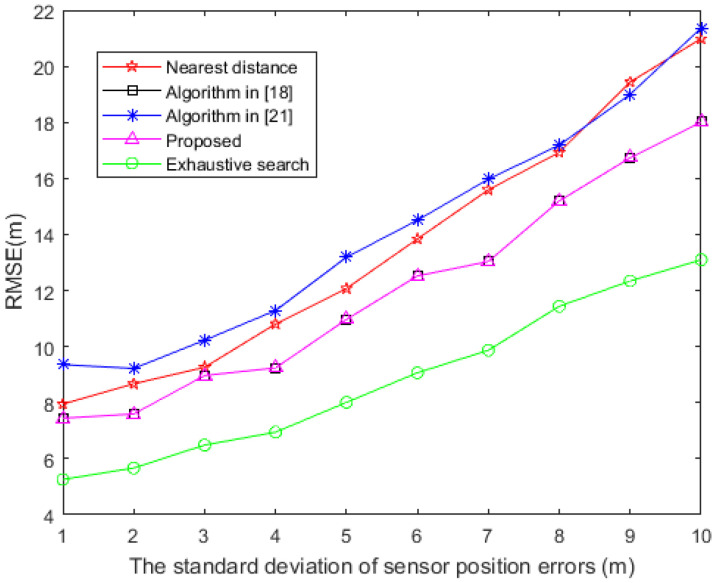
RMSE based on distinct sensor position errors.

**Figure 8 sensors-20-06547-f008:**
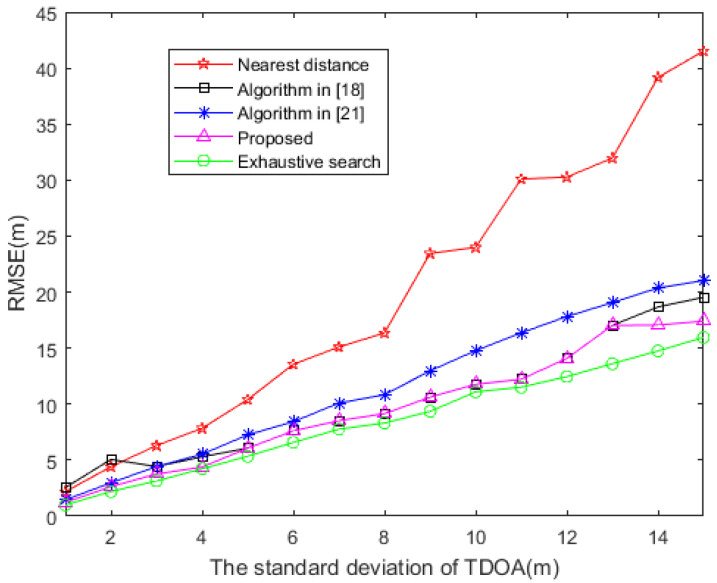
RMSE based on distinct TDOA measurement error (M = 20).

**Figure 9 sensors-20-06547-f009:**
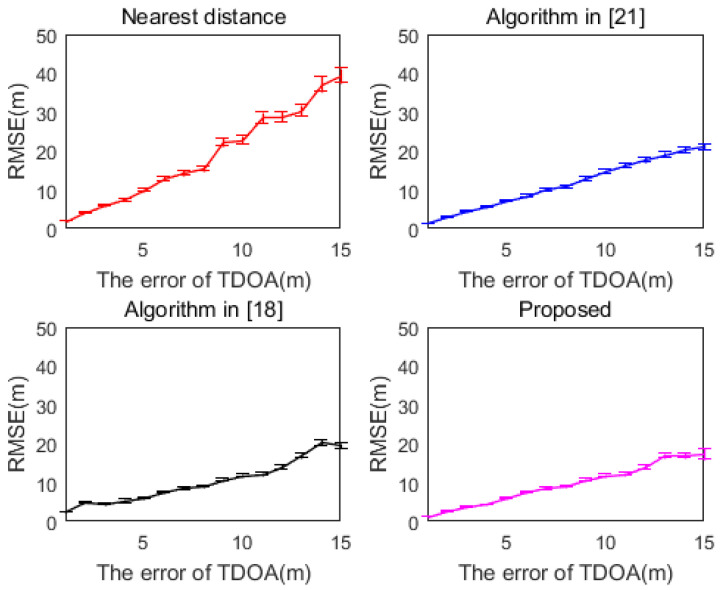
RMSE curve of each method with error bars (M = 20).

**Table 1 sensors-20-06547-t001:** Running time (M = 10).

Algorithm	Average Running Time	RMS Deviation of Running Time
Algorithm in [18]	26.12 s	0.12 s
Algorithm in [21]	25.22 s	0.08 s
Exhaustive search	4.84 s	0.27 s
Proposed	3.02 s	0.02 s

**Table 2 sensors-20-06547-t002:** Running time (M = 20).

Algorithm	Average Running Time	RMS Deviation of Running Time
Algorithm in [18]	32.87 s	0.16 s
Algorithm in [21]	30.57 s	0.09 s
exhaustive search	110.70 s	1.09 s
Proposed	7.44 s	0.05 s

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
