# Peer review of "Time Difference of Arrival Passive Localization Sensor Selection Method Based on Tabu Search"

_sensors, 2020, doi:10.3390/s20226547_

Round 1
Reviewer 1 Report
The manuscript presents a new method for efficiently selecting a subset of measurements (or sensors) in TDOA localization. The authors propose to speed up the sensor selection based on total variance by performing a tabu search in the space of boolean selection vectors. The accuracy of localization results and the runtime are compared with four existing sensor selection methods using simulations.
Main points of criticism:
It is not quite clear to me if the location of the source was varied between simulation runs. If that is the case, all is well, and that should be noted in the introduction to section 4, and the captions of figures 1 and 2 should
mention that the source position is an example only.
If the source position was chosen the same for all simulation runs, that needs to be changed. As both localization accuracy and runtime may depend on the geometry, the source position has to be varied across the whole region covered by the sensors in order to obtain significant results.
The RMS deviation of the runtimes and error bars for the accuracy across the averaged simulation runs should be added to the tables and plots in order to show how much results vary with the source position. This will give an
indication of the robustness of the sensor selection methods.
The nearest distance selection method must be included in the runtime tables. The number of iterations of the tabu search should be given in section 4.
Section 2 follows [22] and [18] very closely. This should be stated clearly at the start of the section.
In (27), the terms not containing any errors are lost after the second line. Is it Delta Omega that is being computed, and subtracting the true value was forgotten in the first two lines? Also, the terms in parentheses like A2 v are column vectors, wich would make Omega a row vector; however, the result of (27) is a column vector, which is inconsistent.
Smaller points:
The English language in the paper is very good. I have, however, a minor quibble with the usage of some technical terms. Terms like "the covariance matrix of the error" or similar are used in the abstract and the text. But what is meant is the covariance matrix of the sensor positions and/or TDOA measurements. The covariance matrix is derived from a set of probability variables, not their errors (though it contains the variances and thereby
indirectly the errors). Similarly, it is stated on p. 6 that the "errors are Gaussian variables", when you mean that the corresonding quantities (positions) have a Gaussian distribution.
Something went wrong with the formulas on page 3: a gap in the right bracket of Delta A2; missing ] in Delta A3 and Delta b; missing equation numbers. The quantities inside Delta(...) should be probability variables notated with a caret ^.
In (25) and in the definition of v, x and y should be x_t and y_t, as before.
Is the minus sign in (33) an oversight? Errors are positive by definition, and the remainder of the expression should already be positive.
In the left part of (36), the subscript of I should be the size of the identity matrix, not a vecctor.
In "author contributions": Supervion -> supervision
Reviewer 2 Report
The authors present an alternative method to solve the sensor selection problem. Based on the experimental results, the method proposed improve others from the literature. However, there are some aspects to take into account:
- some equation numbers are missing
- revise the english style
- missing brackets in equations in page 3
- the presentation of the heuristic or algorithm can be improved
- establish what is the problem of solving the sensor selection problem clearer
- In order to demonstrate the performance of the algorithm proposed the authors should test with other data sets and real scenarios.
Round 2
Reviewer 1 Report
The authors have revised the manuscript in line with most of my suggestions. Eq. (33) has been corrected; I was wrong about Eq. (36); plots with RMS error bars of the positioning accuracy have been added; and many smaller points corrected.
But the main issue remains unchanged. The authors state in their reply that the source position is held constant throughout the simulations performed for the results of each section. This makes the investigation merely anecdotal.
The purpose of a positioning method is to determine the unknown position of an object, the source. This implies that the source position can be anywhere, and an investigation only considering a single position is subject to an
unknown and potentially huge selection bias, however unintentional. The source position should best be varied with a uniform distribution over the whole area in which positioning is to be performed, such as the rectangle displayed in the 2D plots.
This does not mean that the total number of simulation runs needs to increase: Simply choosing a random source position in each run would remedy the issue. Considering the runtime results, rerunning the simulations should be feasible within days with appropriate automation.
It is worth noting that this matter is different from the one investigated in Section 4.2. That section is relevant for the question of how often the selection method would have to be rerun as the source moved, when used in
practice. Varying the source position in the simulations, by contrast, relates to the significance of the results presented in the paper.
Specific points:
The actual number of maximum iterations for the tabu search in the simulations
in sections 4.x is still not given anywhere.
In Fig. 1 the source is plotted at (1200, 1000) instead of (1000, 1200).
Reviewer 2 Report
explain what is passive location
